# Use of Nucleating Agent NA11 in the Preparation of Polyvinylidene Fluoride Dual-Layer Hollow Fiber Membranes

**DOI:** 10.3390/membranes13010075

**Published:** 2023-01-07

**Authors:** Jihyeon Kim, Jinwon Lee, Lindsey B. Bezek, Bumjin Park, Kwan-Soo Lee

**Affiliations:** 1Los Alamos National Laboratory, Los Alamos, NM 87545, USA; 2Department of Materials Engineering and Convergence Technology, Gyeongsang National University, Jinju 52828, Republic of Korea; 3LOTTE Chemical Co., Ltd., Daejeon 305-726, Republic of Korea

**Keywords:** polyvinylidene fluoride (PVDF), dual layer hollow fiber, nucleating agent, thermally induced phase separation (TIPS), non-solvent induced phase separation (NIPS)

## Abstract

Polyvinylidene fluoride (PVDF) dual-layer hollow fiber membranes were simultaneously fabricated by thermally induced phase separation (TIPS) and non-solvent induced phase separation (NIPS) methods using a triple orifice spinneret (TOS) for water treatment application. The support layer was prepared from a TIPS dope solution, which was composed of PVDF, gamma-butyrolactone (GBL), and N-methyl-2-pyrrolidone (NMP). The coating layer was prepared from a NIPS dope solution, which was composed of PVDF, N,N-dimethylacetamide (DMAc), and polyvinylpyrrolidone (PVP). In order to improve the mechanical strength of the dual-layer hollow fiber, a nucleating agent, sodium 2,2′-methylene bis-(4,6-di-tert-butylphenyl) phosphate (NA11), was added to the TIPS dope solution. The performance of the membrane was evaluated by surface and cross-sectional morphology, water flux, mechanical strength, and thermal property. Our results demonstrate that NA11 improved the mechanical strength of the PVDF dual-layer hollow fiber membranes by up to 42%. In addition, the thickness of the coating layer affected the porosity of the membrane and mechanical performance to have high durability in enduring harsh processing conditions.

## 1. Introduction

Polymeric materials have been widely used as membrane materials in the water treatment industry because polymers are generally cheaper than ceramics and metals, and their morphologies and functional properties are easy to control [1]. Recently, polyvinylidenefluoride (PVDF) has received significant attention as a membrane material because it has excellent properties such as good processing, chemical resistance (i.e., chlorine resistance), and thermal and mechanical properties [2,3]. PVDF hollow fiber membranes have been utilized in diverse applications, specifically in industrial wastewater treatment and water purification because of their higher packing density and lower cost compared to flat sheet membranes [4,5,6,7,8,9,10,11,12]. Among the available methods to prepare hollow fiber membranes, thermally induced phase separation (TIPS) is widely used. In general, a homogenous TIPS dope solution is prepared by mixing a polymer with a low molecular weight diluent at a temperature above the polymer’s melting point. To prepare the membranes, this dope solution is extruded through a specifically designed nozzle or cast on non-woven fabric according to the purpose of use. After the coagulation process involving solvent exchange by the diluent in the dope solution, the microporous membrane is formed by phase separation.

TIPS has many advantages over non-solvent-induced phase separation (NIPS) [13,14,15,16,17]. First, the TIPS method has fewer variables to control over the whole procedure. While a NIPS dope solution is composed of a polymer, solvent, and non-solvent, a TIPS dope solution is only composed of a polymer and solvent (or diluent). Second, TIPS can be used for various semi-crystalline polymers that are insoluble in solvents at room temperature. Third, it is possible to significantly increase the polymer content in the TIPS dope solution to improve the mechanical strength of the membrane. Finally, TIPS can more easily create symmetric micro-porous structures, which can be controlled via a solid-liquid (S-L) phase separation or liquid-liquid (L-L) phase separation theory [4,5,18,19]. These theories are important mechanisms for the determination of membrane morphology. While the S-L phase separation is related to polymer crystallization and pores are formed when the diluent is removed, in L-L phase separations, the porous structures are formed mainly by nucleation growth and spinodal decomposition [13,20].

The S-L and L-L phase separation mechanisms are controlled by several conditions such as the polymer-diluent interactions, the polymer concentration in the dope solution, and spinning conditions, among others [6,7,10,13,21,22,23]. Lloyd et al. studied polymer crystallization kinetics and polymer-diluent thermodynamics to determine the mechanism of S-L and L-L phase separations [13,20]. Furthermore, Mingao Gu et al. studied the polymer-diluent interactions as various diluents were used to control polymer concentration and quenching conditions [7,8]. These previous researchers reported that the high polymer concentration, fast cooling rate, and strong interaction of polymer-diluent interaction led to the S-L phase separation in TIPS. By increasing polymer concentration, the TIPS method enables nearly all of the fabricated membranes to have dense top layers and low porosity [10,24,25,26,27,28]. In particular, it was reported that water, among the coagulation solvents, promotes the formation of a dense top layer by causing rapid phase inversion as a strong non-solvent. As a result, some researchers have added organic solvents into a coagulation bath to slow the rate of phase inversion [10,29].

Recently, the triple-orifice spinneret (TOS) was developed to enable co-extrusion with three kinds of dope solutions for preparing hollow fibers. Simultaneous spinning through the TOS could be applied to produce a dual-layer hollow fiber membrane composed of support and coating layers. Further, by adjusting the external dope solution, different morphologies can be prepared for various applications [24,30,31,32,33,34,35]. External dope solutions, such as organic solvents, enable the solvent in the inner dope solution to diffuse out through the external dope solution wall in the coagulation bath. Finally, slow phase separation occurs on the surface of the inner dope solution, and an open porous membrane without a top skin layer can be obtained. Therefore, adjusting the external liquid state through the TOS can enable control of the surface morphology of the membrane [24,30,31].

The addition of additives to a dope solution can also affect the morphology and enhance the membrane properties. Typically, high molecular weight additives are used because they can induce high porosity, create well-interconnected pores, and suppress macro-void formation [36]. Polyvinyl pyrrolidone (PVP) is a common additive owing to its good solubility. PVP is used to make ultrafiltration membranes as a pore-forming agent [31,37]. PVP can also suppress the formation of dense top layers [25]. Other commonly used additives are nucleating agents. The addition of nucleating agents affects the crystallization rate, crystallization temperature, crystal size, and mechanical properties of the polymer [38,39]. In TIPS, polymer crystallization behavior could be an important factor to determine the membrane morphology. In general, the polymer crystallization process mainly depends on the cooling conditions, and in the polymer cooling process, the nucleating agent usually increases the nucleation density, increases the number of spherulites, and decreases the spherulite size. Lloyd et al. studied the addition of nucleating agents to PP-mineral oil systems via TIPS and saw a decrease in spherulite size [13]. Luo et al. studied the effect of diluent types, cooling conditions, extractants, and additive agents (e.g., nucleating agents) on the morphology of TIPS membranes by investigating various microporous polymeric membranes formed via TIPS [40]. Several other papers have reported nucleating agents for PVDF via TIPS [41,42]. However, there is still bias toward the proof of mechanisms and theories, and observing its applicability in water treatment applications has not been investigated to a significant extent.

In this study, porous PVDF dual-layer hollow fiber membranes were fabricated, and the effect of a nucleating agent, NA11, on membrane performance was characterized by the simultaneous spinning of NIPS and TIPS through a self-designed TOS.

TIPS formed a support layer for the dual-layer and NIPS formed a porous coating layer. The performance of fabricated dual-layer membranes of different layer thicknesses was evaluated by morphology, water flux, and mechanical strength. The nucleating agent NA11 was applied to PVDF to improve the mechanical strength of the support layer. The effects of the nucleating agent on PVDF membrane performance were estimated by using scanning electron microscopy (SEM), differential scanning calorimeter (DSC), as well as thermal-optical microscopy. Hence, the controlled pore structure and hierarchical spherulite structure were comprehensively characterized to analyze the effects of the nucleating agent on the membrane pore size and porosity. Furthermore, we optimized the dope solution composition and spinning conditions and evaluated the mechanical properties and water flux compared to single and dual layers of membranes to figure out the best condition for having high water flux and mechanical properties.

## 2. Materials and Methods

### 2.1. Materials

PVDF was purchased from Solvay (Seoul, Republic of Korea)(Solef 1015, Mw = 570,000 g/mol). Gamma-butyrolactone (GBL, diluent), N-methyl-2-pyrrolidone (NMP, solvent), dimethylacetamide (DMAc, solvent), and ethylene glycol (EG, non-solvent) were purchased from the SAMCHUN Chemical (Pyeongtaek, Republic of Korea). NA11 (nucleating agent; NA) was purchased from Asahi Denka (Tokyo, Japan) (ADK STAB NA11, Sodium 2,2’-methylene bis-(4,6-di-tert-butylphenyl) phosphate). PVP (pore-forming additive) was purchased from BASF (Florham Park, NJ, USA) (LUVITEC K-30). The coagulation bath was composed of tap water as the non-solvent. All chemicals were used as received.

### 2.2. Nucleating Agent for PVDF

NA11 is a metal salt of substituted aromatic heterocyclic phosphate (chemical structure shown in Figure 1). It is typically used as a nucleating agent for polyolefins such as polypropylene (PP) and polyethylene (PE) to increase mechanical strength and crystallinity [43,44]. In this experiment, NA11 was dissolved in NMP at 3000 ppm for application to PVDF. The NA11-NMP solution was stirred until it became transparent. The NA11-NMP solution was then placed in the TIPS dope tank and mixed with GBL. The effects of NA11 on PVDF were estimated by measuring the sol-gel transition temperature (T_sg_), thermal properties (by DSC), and isothermal crystallization behavior (by a thermal-optical microscope). The mechanical strength and water flux of the membrane were measured with and without the nucleating agent.

#### 2.2.1. Sol-Gel Transition Temperature (T_sg_)

The T_sg_ of the PVDF-GBL system and the nucleated PVDF-GBL system were measured to determine the effect of the nucleating agent and to determine the spinning temperature of the PVDF-GBL system. Samples were prepared with various PVDF concentrations (25–50 wt.%), GBL (65–45 wt.% by comparing the polymer contents), and either NMP (10 wt.%) (non-nucleated) or NA11-NMP (nucleated), where NA11 (3000 ppm) was dissolved in NMP. The T_sg_ of the PVDF-GBL system was measured by the test tube tilting method. The test tube tilting method measures the temperature where the homogeneous solution does not flow after being kept for 24 h in a convection oven at a set temperature [10,45].

#### 2.2.2. Thermal Properties

The thermal properties were measured using a DSC (TA Instruments Q200). The temperature and heat flow rate were calibrated using indium and tin standards. The membrane samples were prepared by weight (10 mg) and were sealed in the sample pan. The sample pan was preheated at 200 °C for 5 min to remove its thermal history and was cooled to 0 °C at 10 °C/min. Next, the crystallization temperature (T_c_) of the sample was obtained from the peak maximum in the DSC exothermic curve during non-isothermal crystallization. The enthalpy of crystallization (Δ*H_c_*, J/g) was estimated by integrating the exothermic peak. The sample was reheated at 200 °C at 10 °C/min. The melting temperature (T_m_) was obtained from the peak maximum in the DSC endothermic curve. The enthalpy of melting (Δ*H_m_*, J/g) was estimated by integrating the endothermic peak. The degree of crystallinity (*X_c_*) was calculated using:(1)$$X_{c}\left(\%\right)=\left(\Delta H_{c}/105\right)\times 100\%$$

The enthalpy of PVDF with 100% crystallinity was 105 J/g [9]. The crystallization times were also measured. The sample was heated to 200 °C maintained for 5 min and then quenched at 0 °C at a cooling rate of −90 °C/min. The crystallization time was estimated by measuring the time required to reach the DSC exothermic curve peak.

#### 2.2.3. Isothermal Crystallization

The isothermal crystallization of the fabricated membrane was observed by a thermal-optical microscope system. The thermal-optical microscope system was composed of an optical microscope (Olympus, VANOX AHMT3) and a hot stage (LINKAM SCI, THMS600). The prepared sample was pressed at 200 °C to make a 1 mm-thick film. The prepared film was covered with a glass plate and placed on a hot stage at 210 °C for 10 min. Then, it was held at 130 °C, and isothermal crystallization images of the sample were obtained. The isothermal crystallization was captured by a charge-coupled device (CCD) camera, and the spherulite growth was recorded.

### 2.3. Preparation of Dope Solution

In this study, three dope solutions, the TIPS dope solution, NIPS dope solution, and bore solution, were prepared (Table 1). The TIPS dope solution was used to form the support layer, the NIPS dope solution was used to form the coating layer, and the bore solution made the lumen of the hollow fiber. The TIPS dope solution had a high polymer content to improve the mechanical strength of the membrane. PVDF (40 wt.%) was dissolved in GBL (50 wt.%) and NA11-NMP (10 wt.%, 3000 ppm NA11) in the TIPS tank. The prepared solution was heated at 150 °C (30 °C higher than the sol-gel transition temperature). The TIPS tank was agitated for 3 h under nitrogen and then vented for 2 h.

The NIPS dope solution contained a low polymer (PVDF) content and polymer additive (PVP) to form a porous top layer. PVDF (12 wt.%), PVP (10 wt.%), and NMP (72 wt.%) were mixed in the NIPS tank (60 °C, 12 h) under nitrogen. After mixing, the homogeneous NIPS dope solution was kept at room temperature.

The bore solution was made by mixing NMP (solvent) and EG (non-solvent). The ratio of the solvent to non-solvent in the bore solution determines the inner structure of the hollow fiber membrane. A high solvent ratio can produce a loose inner structure. In this study, the ratio of the bore solvent/non-solvent was 3:2.

### 2.4. Preparation of Dual-Layer Hollow Fiber Membrane

The dual-layer hollow fiber membranes were prepared by TIPS and NIPS co-extrusion through a TOS. The TOS was custom-built, and the dimensions are shown in Figure 2. The TIPS dope solution, NIPS dope solution, and bore solution could be simultaneously fabricated for single and dual-layer hollow fibers using the TOS. This TOS was pre-heated and maintained at 140 °C for spinning. Figure 3 shows the spinning system for the dual-layer hollow fiber.

The spinning system was similar to that used in a previous study in a dry-wet spinning system with modified dual-layer spinning [30,46,47]. Three gear pumps were employed to deliver the outer, inner, and bore solutions in the TOS. The gear pump was maintained at 150 °C to produce a suitably-flowing TIPS dope solution. Nitrogen was fed into all spinning dope tanks at 4.0 bar. When the spinning began, the dope and bore solutions were passed through the gear pumps and were delivered to the TOS. The nascent dual-layer hollow fiber membrane was passed through an air gap (4 cm) and was drawn into the coagulation bath. The non-solvent in the coagulation bath was room-temperature tap water. The membranes were formed in the coagulation bath by TIPS and NIPS phase separation. The fabricated membranes were immersed in tap water for 48 h to exchange the water and solvent. After rinsing, the membranes were immersed in a 40 wt.% glycerol/water mixture for 24 h to protect the surface pores. Finally, they were dried at room temperature. All spinning was conducted at room temperature. Table 2 lists the detailed spinning conditions. In addition, to compare the morphology of the single and dual-layer hollow fibers, a single hollow fiber membrane (#1) was prepared by only TIPS. The single-layer membrane was obtained by the same procedure, with exception of the NIPS dope extrusion rate, which was kept at 0 g/min.

### 2.5. Characterization of Dual-Layer Hollow Fiber Membrane

The morphologies of the fabricated membranes were observed by SEM (TOPCON, SM-701). The dried membranes were frozen in liquid nitrogen and were carefully fractured. The prepared samples were sputter-coated with gold (POLARON, SC76102) and the cross-sectional morphology of fibers was observed. In order to observe the fine pores of the sample surface, the samples were sputtered with osmium, and Field emission-SEM (FE-SEM, Hitachi, S-4800 cold type) with high resolution (50,000×) was used to observe the surface.

The water flux of the fabricated membranes was measured by a dead-end type of module. Dead-end type modules were prepared using the membrane sample and a 100 mm-acrylic tube with epoxy potting. One module was potted with 4 membrane fibers. For the water flux test, the modules were pre-wetted with an ethanol/water solution (50 wt.% ethanol) for 3 h. Pure water was fed into the out-in dead-end type modules at 0.1 kg/cm^2^, and the amount of permeated water was calculated. The water flux was calculated by measuring the volume of permeated water per unit time and the unit area of the test membranes:(2)$$\mathrm{Water}\;\mathrm{flux}\;\left(\mathrm{LMH}=L\cdot m^{-2}\cdot {\mathrm{bar}}^{-1}\cdot {\mathrm{hr}}^{-1}\right)=\frac{V}{A\times t}$$
where *V* is the volume of permeate, *A* is the area of the membrane, and *t* is the time of the permeate.

The mechanical properties of the sample fibers were measured using a universal test machine (Instron 5566) with a 50 N load cell. The sample membranes were 3.8 mm in dimension, with a gauge length of 100 mm. The samples were extended by 50 mm/min. The maximum stress data were recorded. Five samples were tested and averaged.

Membrane porosity can be estimated by gravimetrically measuring the fluid displacement and by observing the weight loss of the membrane when immersed in isobutanol, which has low surface tension. First of all, the sample membrane was dried for 5 h at 50 °C and the weight of the dried membrane was measured by a precision balance. Next, the dried membrane was immersed in isobutanol for 24 h and re-weighed after isobutanol was removed from the membrane surface. Finally, the porosity of the membrane, $A_{K}$, was calculated using [6,48]:(3)$$A_{K}=\frac{\left(W_{2}-W_{1}\right)\rho_{1}}{\rho_{1}W_{1}+\left(\rho_{2}-\rho_{1}\right)W_{1}}\times 100\%$$
where *W*_1_ is the initial membrane weight, *W*_2_ is the immersed membrane weight, *ρ*_1_ is the density of PVDF, and *ρ*_2_ is the density of isobutanol.

## 3. Results and Discussion

### 3.1. Effect of Nucleating Agent

#### 3.1.1. Sol-Gel Transition Temperature (T_sg_)

In general, T_sg_ has often been used to indirectly observe the phase diagram of the PVDF-GBL system in the previous study [10,45,49]. Figure 4 shows T_sg_ as a function of polymer concentration in PVDF-GBL and nucleated PVDF-GBL systems. As the PVDF-GBL mixtures showed thermo-reversible properties, they could be gelling at a specific temperature and returning to a solution at a high temperature. However, if diluents such as NMP, DMAc, and Dimethylformamide (DMF) are selected, which have stronger interactions with PVDF, it is difficult to observe the gelation of PVDF-diluent mixtures. Thus, T_sg_ could be observed above the crystallization temperature (T_c_) of the polymer-diluent mixtures. T_sg_ increases gradually with increasing polymer concentration. Higher polymer concentrations in the dope solution usually facilitate the aggregation of the polymer chains and produce a higher crystallinity. Consequently, T_sg_ is related to the crystallization, which increases with increasing polymer concentration.

Nucleation density has been shown to increase with increased polymer concentration [50]. Adding NA11 to the PVDF-GBL system led to similar results, as the T_sg_ of nucleated PVDF-GBL increases with increasing polymer concentration. At higher polymer concentrations, the T_sg_ of nucleated PVDF-GBL is higher than that of the non-nucleated polymer. It can be concluded that the crystallization of PVDF was improved by adding NA11 during cooling and thus the gel network of PVDF-GBL was more pronounced with increasing T_sg_. This also confirms that NA11 is a suitable nucleating agent for the PVDF-GBL system.

#### 3.1.2. Non-Isothermal and Isothermal Crystallization

The thermal behaviors of a neat single PVDF hollow fiber, which will be referred to as #1, and a nucleated single PVDF hollow fiber (#1N) were measured by DSC. From the DSC analysis curves, the melting temperature (T_m_), crystallization temperature (T_c_), enthalpy of the melting (Δ*H_m_*), enthalpy of crystallization (Δ*H_c_*), degree of crystallinity (*X_c_*), and crystallization time were determined. The results are listed in Table 3. The nucleating agent did not have an effect on the T_m_ or Δ*H_m_*. However, the crystallization behavior was considerably different owing to the nucleating agent.

The T_c_ of sample #1 and sample #1N occurred at 134 °C and 142 °C, respectively. The addition of NA11 increased the T_c_ of sample #1N as compared to that of sample #1. Nucleation agents can improve the nucleation efficiency of semi-crystalline polymers by reducing the free-energy barrier so that it increases the crystallization rate and T_c,_ and reduces the crystal size [43,51].

Figure 5 shows DSC exothermic crystallization peaks of samples #1 and #1N during non-isothermal crystallization at a cooling rate of 10 °C/min. The crystallization peak of sample #1N occurred at a higher-temperature region and was narrower than that of #1. As such, DSC confirmed that NA11 serves as a good nucleating agent for PVDF by increasing the T_c_ and reducing the crystallization time. The addition of NA11 results in a higher nucleation density and decreased the chain folding energy to improve crystallization. Furthermore, the narrow DSC peak is attributed to the formation of uniform spherulites owing to the addition of NA11. As NA11 can also act as a catalytic precursor for heterogeneous nucleation, the crystallization rate of #1N may be faster. As confirmed in Table 3, sample #1N shows a higher *X_c_* and reduced crystallization time as compared to sample #1.

The effect of added NA11 can also be observed during isothermal crystallization. The spherulite growth of samples #1 and #1N was examined at 130 °C via a thermal-optical microscope in Figure 6. The melted samples were quenched at 130 °C, and the photos were taken after the isothermal crystallization was complete. As shown in Figure 6a, sample #1 exhibited large spherulites after 5 min, whereas for sample #1N, no additional crystal growth was detected after 5 min in Figure 6b, so the spherulites were very fine.

Owing to the addition of NA11, the crystallization of #1N might be initiated via heterogeneous nucleation. As sample #1N had a greater number of nuclei and spherulites than sample #1, the spherulite growth may have been suppressed by impinging each other.

### 3.2. Single Layer Membrane—Morphology

Figure 7a–c shows the cross-sectional structures of sample #1, and Figure 7d shows the outer surface of sample #1. The outer diameter and thickness were about 1200 μm and 200 μm, respectively. The single-layer membrane was obtained with a TIPS extrusion rate of 17 g/min and a bore extrusion rate of 6 g/min. Sample #1 exhibits a dense top layer and interconnected spherulite structures on the sub-layer (Figure 7b,c). Surface pores are not observed due to the dense top layer [10].

In general, the morphology of the semi-crystalline polymer membrane can be affected by the cooling rate. In this experiment, the dense top layer can be formed by a rapid cooling rate. When the hot dope solution from the nozzle is immersed in the coagulation bath, the phase separation of the hot dope solution occurs quickly due to loss of thermal energy. Since the outer dope solution is directly in contact with the water, the cooling rate of the surface layer is the fastest. As confirmed in this experiment, a fast-cooling rate increases the polymer crystallization rate and forms very fine crystals. Thus, these fine crystal spherulites could form the dense surface.

Another possible reason for the formation of the dense top layer is the selection of water as the non-solvent. Water induces fast precipitation of the dope solution [30]. The precipitation rate usually can be determined by the interactions between the solvent and non-solvent. Organic solvents such as NMP, DMF, and DMAc have good interaction with water owing to their carboxyl groups that can form hydrogen bonds with water. As GBL also has a carboxyl group, when the PVDF-GBL dope solution is immersed into the coagulation bath, the solvent in the dope solution rapidly outflows into the coagulation bath because the dope solution has a higher diffusion rate than that of the bath. This increases the polymer concentration on the surface layer and could cause polymer aggregation and crystallization.

The cross-sectional sub-layer morphology of the single-layer membrane (Figure 7b,c) exhibits interconnected spherulites that are 1–5 μm. This morphology is attributed to the S-L phase separation during the cooling process. This result is similar to the literature previously reported regarding the PVDF-GBL system [9,10,45,49]. The spherulites and granules can be observed in the S-L phase separation, which is related to polymer crystallization. In general, the morphology of a membrane is determined by the polymer concentration, polymer-diluent interactions, and cooling rate [52]. In this study, the S-L phase separation was mainly observed by the following experimental conditions:

First, the PVDF concentration in the TIPS dope solution was fixed at 40 wt.% in order to obtain a high mechanical strength; this spinning condition might affect the membrane morphology. Increasing polymer concentrations in dope solutions lead to S-L phase separation in TIPS [6]. The higher polymer content of the dope solution causes the polymer chains to aggregate and increases the crystallinity during the phase separation [7]. Consequently, as the higher polymer content affects the phase separation mechanism, the S-L separation was dominant.

Second, the interactions between the polymer and diluent affect the morphology of the membrane [53]. In this study, GBL was used as the diluent for PVDF, and PVDF exhibited good compatibility with GBL due to dipole-dipole interactions between the fluorine of PVDF and the carboxyl group of GBL. When the polymer is dissolved in a suitable solvent, the polymer chains have good mobility and are easy to aggregate and thus improving the crystallization during cooling [6]. Similarly, in the Flory theory of melting point depression, the melting point of polymers decreases in suitable solvents. Although melting point depression cannot truly be compared to crystallization temperature, the crystallization temperature of a polymer in a good solvent can be lower than that of a polymer in a poor solvent [54]. Suitable interactions of PVDF-GBL led to good miscibility and reinforced the polymer crystallization during cooling, but also increased the occurrence of the S-L phase separation.

Third, the quenching conditions affect the morphology of the membrane during TIPS [55]. The quenching condition was controlled by the difference between the coagulation bath temperature and the dope solution temperature. In general, by increasing the temperature gap, the cooling rate of the dope solution is increased during cooling. With a rapid cooling rate, S-L phase separation occurs prior to L-L phase separation. In this study, the coagulation temperature and dope solution temperature were fixed at 25 °C and 150 °C. The morphologies of all fabricated membranes show interconnected spherulites, which are attributed to the quenching conditions. The spherulites shown in Figure 7b,c are formed by the crystallization process through the S-L phase separation. In addition, as the polymer crystallized, the diluent was rejected from the inter-lamellae and inter-spherulite regions [6,7,9]. The rejected diluent accumulated in the inter-spherulite region and remained in the inter-lamellae. Consequently, the diluent was extracted during the cleaning process and the inter-lamellae and inter-spherulite regions formed the inner pores of the membrane.

### 3.3. Dual-Layer Membrane- Morphology

Figure 8 shows the prepared dual-layer hollow fiber membrane, which will be referred to as sample #2. The membrane is composed of a support layer (the same TIPS layer as used for sample #1) as well as a coating layer formed by NIPS (composed of PVDF (12 wt.%), PVP (10 wt.%), and NMP (78 wt.%)).

Figure 8c shows the cross-sectional morphology of the support layer from the TIPS process for sample #2, which resembles that of sample #1. However, in contrast to sample #1, the cross-sectional morphology of the coating layer exhibits finger-like forms, as highlighted in Figure 8b, and surface pores, as observed in Figure 8d. The NIPS coating layer plays an important role in making the porous surface on the TIPS support layer. During the simultaneous spinning process, the NIPS dope solution entirely covers the TIPS dope solution as soon as it is extruded through the nozzle of the TOS. Since the NIPS dope solution has plenty of organic solvents, it acts as the first coagulation bath for the TIPS dope solution. This NIPS coating barrier could prevent solvent evaporation of the TIPS solution at the air gap state and prevent the dope solution from rapidly spreading into a coagulation bath. Therefore, the interface polymer concentration of the TIPS dope solution slowly increases by the NIPS coating barrier in the phase separation system. As a result, as shown in Figure 8a,b, the dense top layer was not observed at the surface and interface of the dual layer; instead, a porous PVDF membrane is obtained by using the TOS and NIPS coating layers.

Finger-like structures were observed on the coating layer in Figure 8b, which are typically formed by instantaneous de-mixing via the rapid exchange of the solvent and non-solvent [56]. Instantaneous de-mixing occurs via strong interactions between the solvent and non-solvent. Because water is a strong non-solvent and is forming hydrogen bonds with NMP, instantaneous de-mixing occurs due to the fast exchange of the non-solvent and solvent. Liquid-liquid de-mixing also occurs at the interfacial surface when the NIPS dope solution contacts the non-solvent in the bath. Consequently, a skin layer is formed on the coating layer, as opposed to the dense top layer. Surface pores are formed on this skin layer by the addition of a pore-forming agent, PVP, which has previously been used as a successful pore-forming agent [57]. Formed pores were observed at about 0.032 μm on the surface of sample #2 in Figure 8d.

### 3.4. Dual-Layer Membrane—Effect of NA11 on Crystallization

The dual hollow fiber membrane samples (#2 and #2N) were fabricated to observe the effects of the nucleating agent NA11 in the TIPS dope solution. Figure 9 shows the cross-sectional morphology of samples #2 and #2N. For these samples, an interconnected spherulite structure is observed, potentially owing to the S-L phase separation during the cooling process. According to the previous results, it was predicted that the nucleating agent would change the membrane morphology. Although we confirmed that NA11 decreased the spherulite size during PVDF isothermal crystallization, there was no noticeable difference in the membrane morphologies. The samples instead exhibited similar spherulite sizes (approximately 1 to 5 μm; in Figure 9).

It was previously reported that the crystallization rate of semi-crystalline polymers increases with increased cooling rates [58]. Su et al. also reported that high cooling rates decrease the spherulite size in the membrane of the PVDF-GBL system during TIPS [8]. In this study, the hot dope solution (150 °C) was immersed in the cool coagulation bath (25 °C) for the phase separation. As such, the cooling rate of the dope solution can affect the morphology of the membrane. This quenching condition may affect the morphology of the membrane more than the nucleating agent. Although the nucleating agent did not have an effect on the membrane morphology, the mechanical properties of the fabricated membrane were affected by the addition of NA11.

### 3.5. Effect of NA11 on Mechanical Properties

The mechanical strength and flux of the fabricated single and dual-layer hollow fiber membranes are shown in Figure 10. The addition of NA11 significantly improves the mechanical strength of single and dual-layer hollow fiber membranes (#1N and #2N), as seen in Figure 10a. In particular, the mechanical strength of sample #1N increases by about 41% with the addition of NA11 due to the dense top layer, where NA11 improves the gelation and crystallization of the polymer upon cooling.

The single-layer membranes (#1, #1N) showed zero water flux due to the dense top layer, as shown in Figure 10b. As confirmed in Figure 7d, there were no pores on the surface of the single-layer membrane due to the dense top layer. This could make it difficult to apply sample #1 in the water treatment industry [59].

The mechanical strength and water flux of samples #2 and #2N can also be found in Figure 10. Although these samples show lower mechanical strength than samples #1 and #1N, they show higher water flux owing to the difference in their morphologies: specifically, the porous structure and coating layer. The dual-layer samples (#2, #2N) have water flux above 500 LMH, without a dense top layer, and exhibit lower mechanical strength. In the case of sample #2N, they have a minor reduced water flux due to improving crystallization by NA11. On the other hand, the mechanical strength of sample #2N shows slight improvements relative to sample #2. In general, commercial microporous membranes for water treatment have 27 to 313 LMH for water flux [60]. Therefore, our developed membrane has a high potential for improving efficiency operation.

### 3.6. Effect of Membrane Thickness on Performance

Dual-layer hollow fiber membranes with various thicknesses were prepared by controlling the extrusion of TIPS and NIPS to estimate the membrane performance. The thickness of the support and coating layers could be determined by controlling the extrusion rate of TIPS and NIPS in this experiment.

Figure 11a shows the variation of mechanical strength and water flux of the dual-layer hollow fiber membrane when varying the TIPS dope extrusion rate. The extrusion rates of the NIPS dope solution and bore solution were fixed at 7 g/min and 6 g/min, respectively. As the TIPS dope extrusion rate increases, the thickness of the support layer increases. As the result, the mechanical strength of the membrane increases, and the water flux decreases. In the previous morphology analysis of the dual-layer membrane, the support layer from TIPS showed a denser structure than the coating layer from NIPS. Therefore, it follows that increasing the extrusion of TIPS, and thus the thickness of the support layer, can reduce water flux and improve mechanical strength.

On the other hand, adjusting the extrusion rate of the NIPS dope solution has an opposite effect compared to that of the TIPS dope solution. Even though the membrane size increases when the NIPS extrusion rate increases, the dual-layer membranes exhibit an increased water flux and decreased mechanical strength, as shown in Figure 11b. The TIPS dope extrusion rate and bore extrusion rate was fixed at 17 g/min and 6 g/min, respectively.

Additional experiments were conducted as follows in order to understand the results in Figure 11b. Figure 12 shows the cross-sectional morphologies of the dual-layer membrane with various NIPS dope solution extrusion rates. As the extrusion rate of the NIPS dope solution increases, the coating layer thickness gradually increases. These coating layers exhibit macro voids within a finger-like structure, as seen in Figure 12. The increasing water flux in Figure 11b can be attributed to the thickening porous, coating layer.

Figure 13 shows the porosity measurements for all membranes in Figure 12. The porosity increases linearly with increasing NIPS extrusion rate. In general, the higher the porosity of the membrane, the lower the mechanical strength [4,9,61]. This explains why the mechanical strength decreases with increasing NIPS extrusion rate as shown in Figure 11b.

In this dual-layer membrane system, the thickness of the coating layer is an important factor in determining the porosity of the prepared membrane. The membrane porosity mainly affected the performance, water flux, and mechanical strength, in this dual-layer system. In Figure 11b, we could say that the condition of extrusion rate of TIPS 17 g/min and NIPS 7 g/min showed the best membrane performance with a good balance between water flux and mechanical properties in our experiment. Moreover, we can use this fabrication method to tailor thickness to our needs.

## 4. Conclusions

PVDF dual-layer hollow fiber membranes were successfully prepared by TIPS and NIPS through a TOS. The NIPS dope solution was applied as the first coagulation bath for the TIPS dope solution. It suppressed the formation of the dense top layer on the supporting layer and formed a porous coating layer with finger-like structures. The TIPS dope solution was simultaneously spun with the NIPS and bore dope solutions to form the support layer. The morphology of the support layer was an interconnected spherulite structure that is related to S-L phase separation. The prepared PVDF dual-layer membrane showed a water flux and tensile strength of 560 LMH and 5.1 MPa, respectively.

To improve the mechanical strength of the fabricated PVDF membrane, the nucleating agent NA11 was applied. NA11 served as an effective nucleating agent for PVDF by increasing the T_sg_, T_c_, and crystallization rate. As a result of the addition of NA11, the tensile strength of the fabricated membrane was improved by increasing the crystallinity of the polymer. In particular, the tensile strength of the single-layer membrane was increased by about 12 MPa owing to the reinforced dense top layer, but there was no water flux. Although the nucleated dual-layer membrane (#2N) exhibited a slightly improved mechanical strength of about 6.2 MPa as compared to the net dual-layer membrane (#2), lower water flux was measured at about 540 LMH.

The dual-layer hollow fiber membranes with various thicknesses of coating and support layers were also prepared and tested. The performance of the membranes was mainly dependent on the thickness of the coating layer, which was dictated by the TIPS and NIPS extrusion rates. The coating layer, which has macro voids within the finger-like structures, determined the porosity of the membrane. Therefore, we have observed that increasing the thickness of the coating layer increased water flux and decreased the mechanical strength of the membrane.

## Figures and Tables

**Figure 1 membranes-13-00075-f001:**
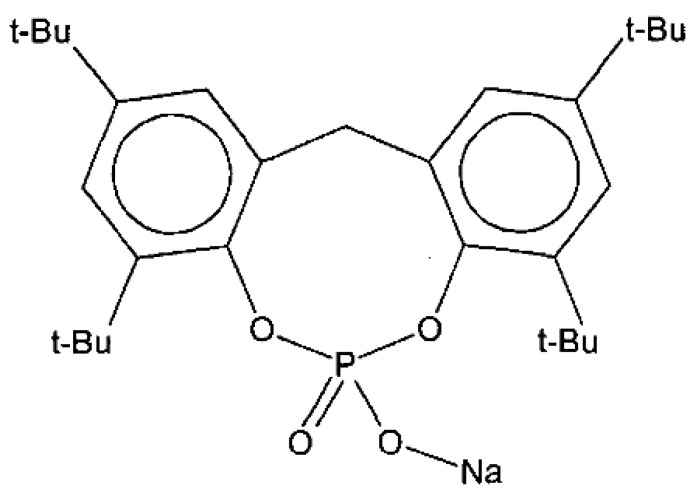
Chemical structure of nucleating agent NA11.

**Figure 2 membranes-13-00075-f002:**
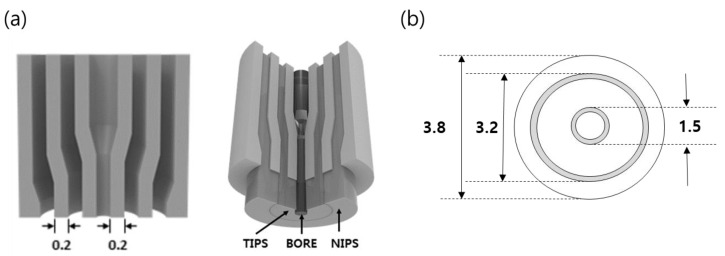
Schematic diagram of the triple orifice spinneret (TOS): (**a**) Cross section of the TOS. The dope solutions are pumped simultaneously to three line of the TOS, (**b**) Dimension of the TOS (unit: mm).

**Figure 3 membranes-13-00075-f003:**
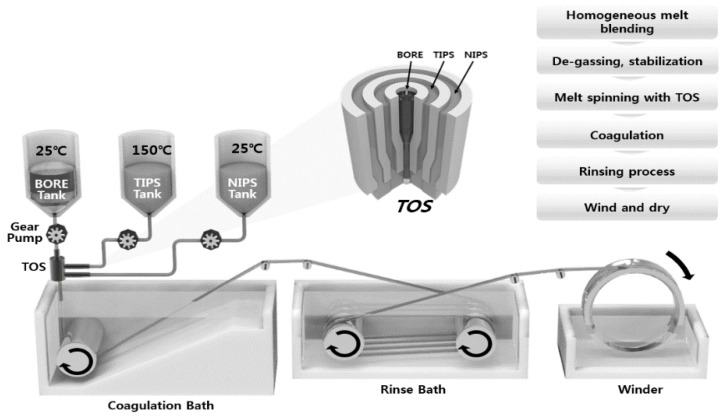
Schematic diagram of TIPS and NIPS simultaneous spinning machine.

**Figure 4 membranes-13-00075-f004:**
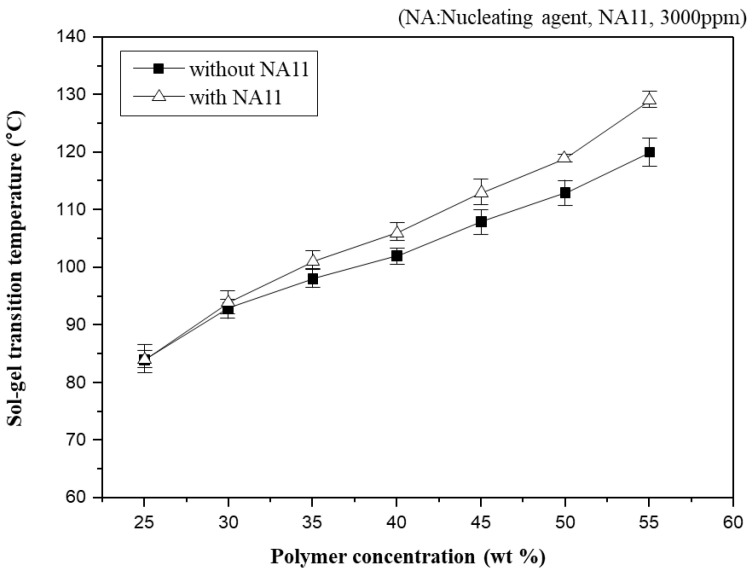
Sol-gel transition temperature changes (T_sg_) as a function of polymer concentration for the PVDF/GBL system.

**Figure 5 membranes-13-00075-f005:**
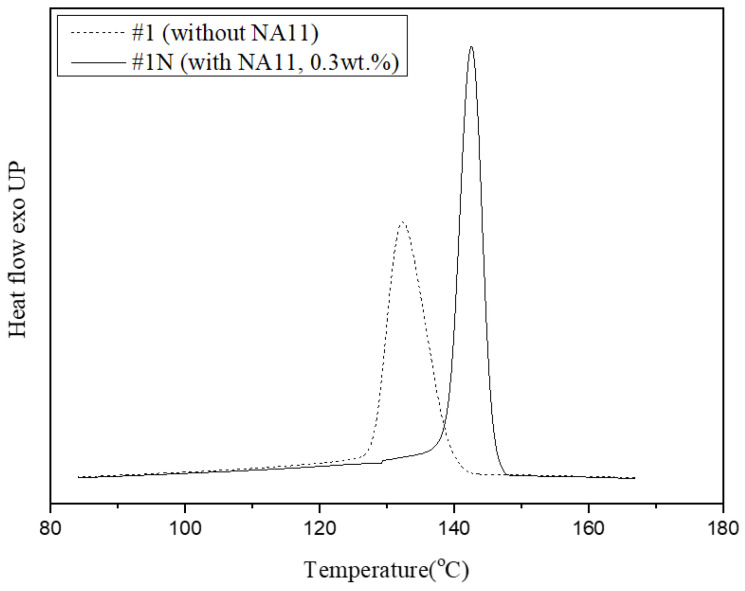
Non-isothermal crystallization peak shifts of PVDF with and without the nucleating agent at 10 °C/min.

**Figure 6 membranes-13-00075-f006:**
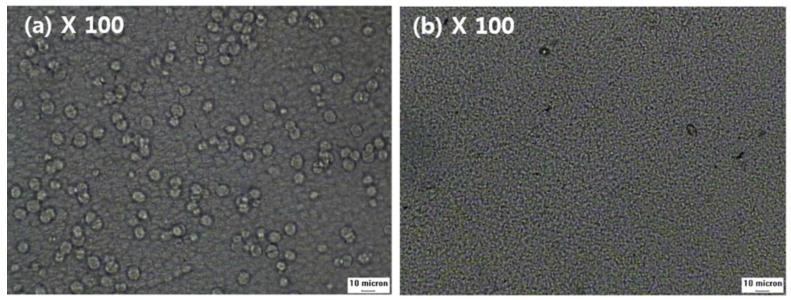
Thermal-optical microscope images of PVDF with and without NA11 during isothermal crystallization: (**a**) The image of sample #1 after 5 min at 130 °C, (**b**) The image of sample #1N after 5 min at 130 °C.

**Figure 7 membranes-13-00075-f007:**
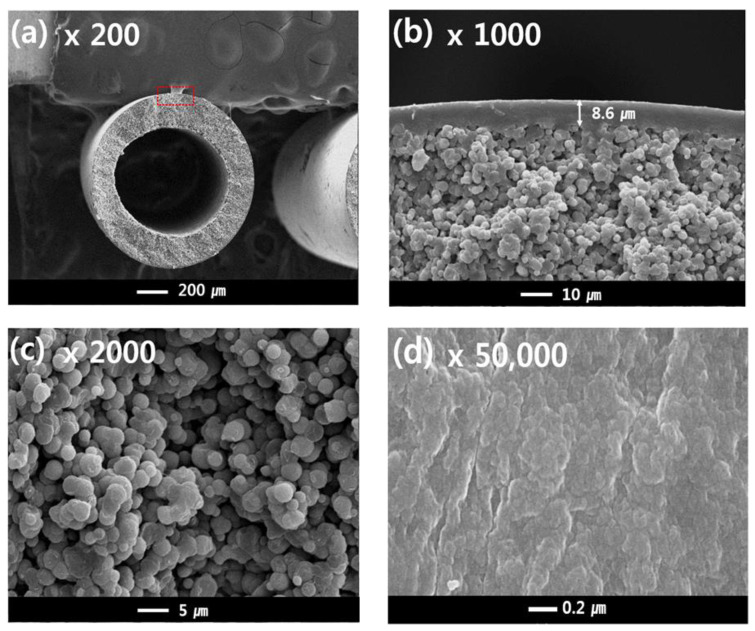
The morphology of the PVDF single layer hollow fiber membrane: cross-sectional images are at (**a**) 200×, (**b**) 1000×, and (**c**) 2000× magnification; (**d**) surface image at 50,000× magnification; Hollow fiber spinning conditions: Sample #1, TIPS = 17 g/min, NIPS = 0 g/min, Bore = 6 g/min.

**Figure 8 membranes-13-00075-f008:**
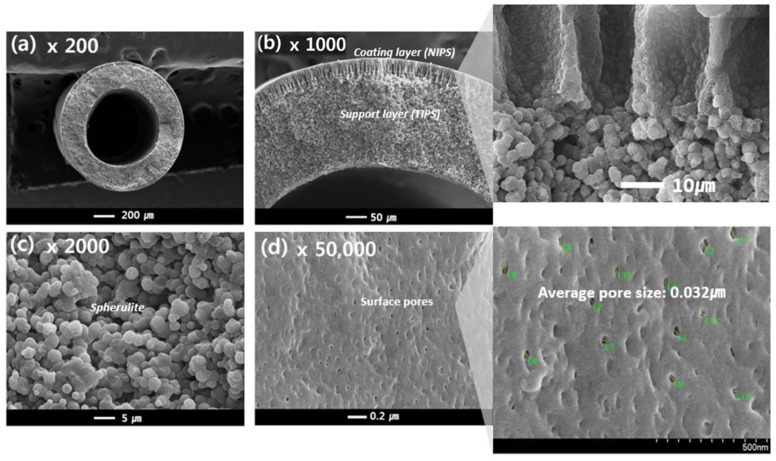
The morphology of the PVDF dual-layer hollow fiber membrane: Cross-sectional image: (**a**) 200×, (**b**) 1000× (highlight 1800×), (**c**) 2000×, and Surface image: (**d**) 50,000× (highlight 70,000×) at magnification; Hollow fiber spinning conditions: Sample #2, TIPS = 17 g/min, NIPS = 7 g/min, Bore = 6 g/min.

**Figure 9 membranes-13-00075-f009:**
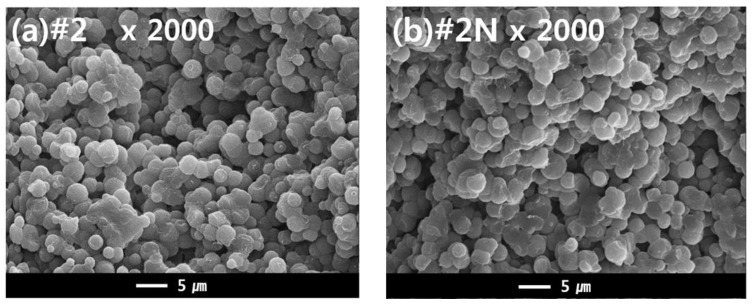
SEM images comparing the cross-sectional morphologies of dual-layer PVDF membranes (1) without, and (2) with the nucleating agent. Hollow fiber spinning conditions: TIPS = 17 g/min, NIPS = 7 g/min, Bore = 6 g/min. (**a**) Cross sectional image of Sample #2 at ×2000, and (**b**) Cross sectional morphology of sample #2N (nucleated) at ×2000.

**Figure 10 membranes-13-00075-f010:**
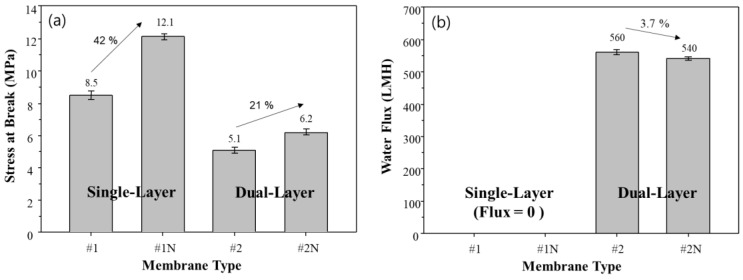
Effect of added nucleating agent on membrane performances. (**a**) Tensile strength, and (**b**) Water flux of single and dual-layer hollow fiber membranes.

**Figure 11 membranes-13-00075-f011:**
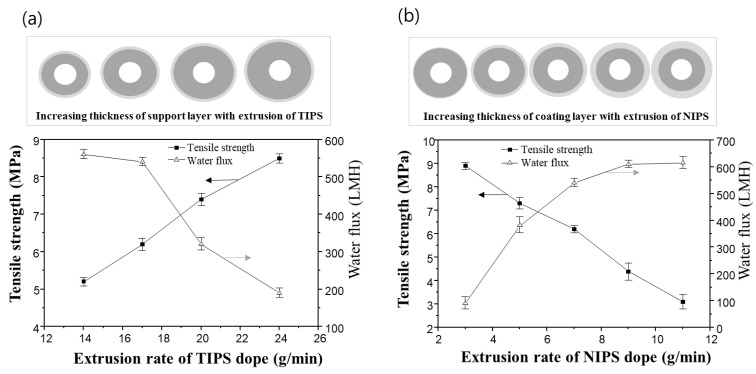
The tensile strength and water flux changes for sample #2N in the function of the extrusion rate of TIPS and NIPS dope, respectively: (**a**) the variation of tensile strength and water flux via TIPS extrusion rate. Fixed at NIPS extrusion rate 7 g/min and bore extrusion rate 6 g/min, and (**b**) the variation of tensile strength and water flux via NIPS extrusion rate. Fixed at TIPS extrusion rate 17 g/min and bore extrusion rate 6 g/min.

**Figure 12 membranes-13-00075-f012:**
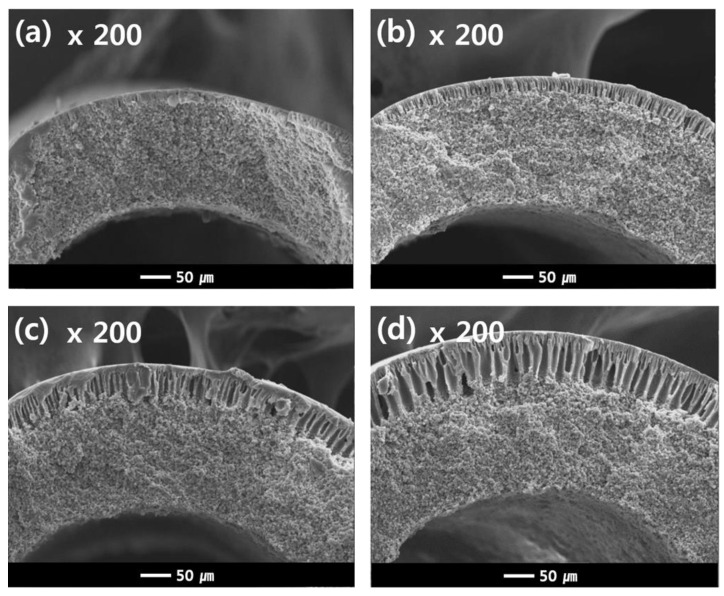
Cross-sectional SEM images of the dual-layer hollow fibers (Sample #2N) as a function of NIPS extrusion rate. The NIPS extrusion rate is (**a**) 3 g/min (**b**) 5 g/min (**c**) 7 g/min and (**d**) 9 g/min, the TIPS extrusion rate is 17 g/min, and the bore extrusion rate is 6 g/min.

**Figure 13 membranes-13-00075-f013:**
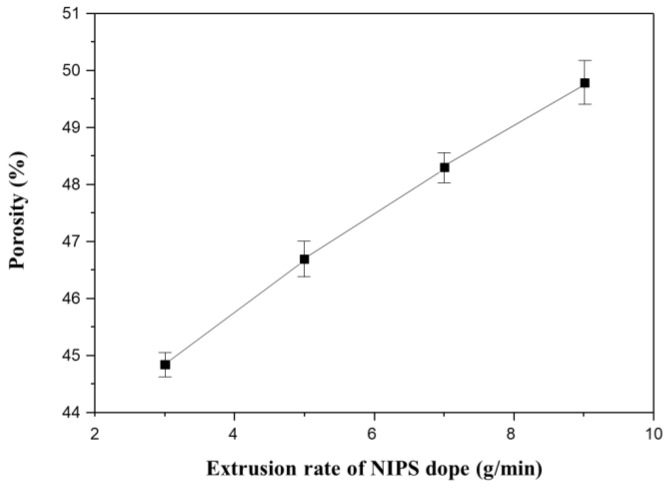
The variation of porosity as a function of extrusion rate of NIPS dope solution in the dual layer hollow fiber (Sample 2#N): Fixed at TIPS extrusion rate 17 g/min and at bore extrusion rate 6 g/min.

**Table 1 membranes-13-00075-t001:** Dope solution composition of the single and dual layer membrane.

Sample #	Membrane Type	TIPS Dope Solution (wt.%)	NIPS Dope Solution (wt.%)	Bore Solution (wt.%)
#1	Single layer	PVDF/NMP/GBL = 40/10/50	-	NMP/EG = 60/40
#1N	Single layer + NA11	PVDF/NMP/GBL/NA11 = 40/10/50/0.3	-	NMP/EG = 60/40
#2	Dual layer	PVDF/NMP/GBL = 40/10/50	PVDF/NMP/PVP = 12/78/10	NMP/EG = 60/40
#2N	Dual layer + NA11	PVDF/NMP/GBL/NA11 = 40/10/50/0.3	PVDF/NMP/PVP = 12/78/10	NMP/EG = 60/40

**Table 2 membranes-13-00075-t002:** Various spinning conditions of single and dual-layer hollow fiber membranes.

Conditions	TIPS Tank	NIPS Tank	Bore Tank
Mixing temperature	150 °C	25 °C	25 °C
Heating line temperature	150 °C	25 °C	25 °C
Mixing time	Mixing for 3 h Venting for 2 h	12 h	0.1 h
Nitrogen pressure	4 bar	4 bar	-
Extrusion rate by gear pump	14–24 g/min	3–11 g/min	6 g/min
Nozzle temperature	140 ℃		
Air gap	4 cm		
Coagulation bath temperature	25 ℃		
Rinsing bath temperature	25 ℃		
Extrusion rate	3 m/min		

**Table 3 membranes-13-00075-t003:** Thermal properties of PVDF membranes with and without the nucleating agent.

Sample	T_m_ (°C)	T_c_ (°C)	Δ*H_m_* (J/g)	Δ*H_c_* (J/g)	*X_c_* (%)	T_c_ Time (s)
Sample #1 (Neat PVDF)	170	134	54	49	47	77
Sample #1N (NA11 0.3 wt.%)	171	142	55	54	51	68

## Data Availability

Data sharing not applicable.

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
