# Peer review of "Use of Nucleating Agent NA11 in the Preparation of Polyvinylidene Fluoride Dual-Layer Hollow Fiber Membranes"

_membranes, 2023, doi:10.3390/membranes13010075_

Round 1
Reviewer 1 Report
The paper deals with the dual-layer hollow fiber membrane preparation for water treatment application. The improvement of mechanical strength of the dual-layer hollow fiber membrane owing to the nucleating agent NA11 presence. It is interesting paper but needs some improvements, as below described, before its publication.
1] What is the morphology of single layer membrane with NA11.
2] Whether the doping amount of NA11 has any effect on the membrane structure and performance.
3] The title of the manuscript emphasizes the use of NA11, but the morphology of the membrane is almost all without NA11, especially the surface morphology. Why?
4] According to Figure 8, the surface morphology or surface porosity of membranes have a great influence on the water flux. Because the pore size of the surface is significantly smaller than that of the finger-like pore, and the surface porosity is obviously not high.
5] In the text (lines 554-556) it is mentioned that the condition of extrusion rate of TIPS 17g/min and NIPS 6g/min showed the best membrane performance with a good balance between water flux and mechanical properties in our experiment. This is correct according to the trend of the curve. However, no data of membrane properties is obtained under the condition that the extrusion rate of NIPS is 6g/min. I think that the data of the water flux and mechanical strength should be given for this condition.
Author Response
To Reviewer #1
The paper deals with the dual-layer hollow fiber membrane preparation for water treatment application. The improvement of mechanical strength of the dual-layer hollow fiber membrane owing to the nucleating agent NA11 presence. It is interesting paper but needs some improvements, as below described, before its publication.
We really appreciate the time and effort that you have dedicated to providing your valuable feedback on our manuscript. We have carefully addressed all the comments. The corresponding changes and refinements made in the revised paper are summarized in our response below. We have highlighted the changes within the manuscript.
1] What is the morphology of single layer membrane with NA11.
A1] Thank you for the valuable comments. The most important thing in this study is to figure out and compare the cross-sectional morphologies and their performance of the membranes each other when we prepared the dual layer with the TOS.
As you know, the morphologies (even performance) of membranes are totally different based on the preparation methodology with lots of parameters and conditions (e.g. temperature, bore solution, cooling time/temperature, spinning rate, dope solvent, etc). In other words, when we prepare the hollow fiber even by using the same polymer material for a single layer and/or a dual layer, we should have to use completely different preparation parameters. Therefore, instead of comparing morphologies of a single layer with and without NA11, we would like to focus on the mechanical properties of the supporting layer (PVDF/NA11) using the TIPS method and the synergistic effect of porous structure, dense top layer prepared by the NIPS method, on the supporting layer prepared by using TOS at the same time.
As a result, as you can see the Figure 9, we couldn’t find any differences between the morphologies of the membrane with and without NA11. Here in this study, we have focused on preparing the dual layer and the effect of NA11, mostly mechanical strength, on the performance of the dual layer.
2] Whether the doping amount of NA11 has any effect on the membrane structure and performance.
A2] We haven’t investigated the effect of NA11 loading on the properties and/or performances. As we stated in our manuscript, NA11 is typically used as a nucleating agent for polyolefins to increase mechanical strength and crystallinity (ref#43, 44). As our initial study, we would like to check the possibility of NA11 in PVDF if it is applicable to dual layer membranes. As a future study, we plan to understand the loading effect on the structure and performance.
3] The title of the manuscript emphasizes the use of NA11, but the morphology of the membrane is almost all without NA11, especially the surface morphology. Why?
A3] As shown in Figure 9, we couldn’t find any specific difference in the morphologies of both membranes (with and without NA11). Both morphologies are shown similarly in Figures 7 and 8.
4] According to Figure 8, the surface morphology or surface porosity of membranes have a great influence on the water flux. Because the pore size of the surface is significantly smaller than that of the finger-like pore, and the surface porosity is obviously not high.
A4] Yes, as you address, we have pretty good water flux even with smaller surface pore sizes. That is because there are not much of dead-end pores in the hollow fiber as shown in Figure 12 (cross-sectional image).
5] In the text (lines 554-556) it is mentioned that the condition of extrusion rate of TIPS 17g/min and NIPS 6g/min showed the best membrane performance with a good balance between water flux and mechanical properties in our experiment. This is correct according to the trend of the curve. However, no data of membrane properties is obtained under the condition that the extrusion rate of NIPS is 6g/min. I think that the data of the water flux and mechanical strength should be given for this condition.
A5] That is a typo, so the rate should be 7g/min for NIPS. Thank you for pointing it out. We changed the typo in the manuscript.
Reviewer 2 Report
I think it is of great interest in the community of membrane preprations. As a result, I will recommend the publication of this manuscript to accept it followed by some minor corrections.
Comments.
1. Please check English in overall manuscript.
2. Please include the AFM for surface roughness.
3. I will appreciate if author can explain that how they have calculated the Tm in table 3.
4. In Figure 13 is author have checked doping more than 9 g/min?
5. Please make same format for all the references. Reference 58, 59, 60 have stop signs after journal name whereas reference 61 have comma after journal name.
Author Response
I think it is of great interest in the community of membrane preprations. As a result, I will recommend the publication of this manuscript to accept it followed by some minor corrections.
We really appreciate the time and effort that you have dedicated to providing your valuable feedback on our manuscript. We have carefully addressed all the comments. The corresponding changes and refinements made in the revised paper are summarized in our response below. We have highlighted the changes within the manuscript.
1] Please check English in overall manuscript.
A1] We have checked English in this manuscript twice. We think this revised version is good enough for readers to read our manuscript.
2] Please include the AFM for surface roughness.
A2] Thank you for pointing this out. Surface roughness is one of the important factors to understand the effect of bio-fouling on the membrane performance. Although we agree that this is an important consideration, it cannot be analyzed in this manuscript because we do not have any more samples for the AFM study. As a future study, we have a plan to understand “The effect of NA11 loading on the structure and performance”, including the surface roughness and morphologies through the AFM study.
3] I will appreciate if author can explain that how they have calculated the Tm in table 3.
A3] We haven’t calculated Tms of samples. In this study, Tms of the samples in Table 3 were measured by DSC.
4] In Figure 13 is author have checked doping more than 9 g/min?
A4] 9 g/min is the max rate we have used for this study.
5] Please make same format for all the references. Reference 58, 59, 60 have stop signs after journal name whereas reference 61 have comma after journal name.
A5] Thank you for pointing it out. We have changed into the correct reference style. In addition to the above comments, all spelling and grammatical errors pointed out by the reviewers have been corrected in revised manuscript.